# Absence of Association between Child Temperament and Early Childhood Caries: A Cross-Sectional Study

**DOI:** 10.3390/ijerph20043251

**Published:** 2023-02-13

**Authors:** Rodrigo Mariño, Paulina Hofer-Durán, Javiera Nuñez-Contreras, Yanela Aravena-Rivas, Carlos Zaror

**Affiliations:** 1Melbourne Dental School, University of Melbourne, Melbourne 3010, Australia; 2Dental School, Faculty of Dentistry, Universidad de La Frontera, Temuco 4781176, Chile; 3Programa de Magister en Odontología, Facultad de Odontología, Universidad de La Frontera, Temuco 4811230, Chile; 4Department of Pediatric Dentistry and Orthodontics, Faculty of Dentistry, Universidad de La Frontera, Temuco 4781176, Chile; 5Center for Research in Epidemiology, Economics and Oral Public Health (CIEESPO), Faculty of Dentistry, Universidad de La Frontera, Temuco 4811230, Chile

**Keywords:** early childhood caries, child temperament, Chile

## Abstract

Early childhood caries (ECC) is a worldwide public health problem. The biological and behavioural determinants that are directly involved in ECC have been well documented; however, evidence on the effects of some psychosocial factors remains conflicting. This study aimed to assess the association between child temperament and ECC in Chilean preschoolers. Prior approval of the protocol was obtained from the ethics committee of Universidad de La Frontera (Folio N° 020_17), and all of those involved in the study provided signed informed consent forms. The cross-sectional study was conducted with 172 children aged 3 to 5 years attending preschools in Temuco, Chile. Each child’s temperament was assessed based on parents’ responses to the Early Childhood Behaviour Questionnaire. The outcomes assessed were caries prevalence and caries experience (dmft scores). The covariates included were socioeconomic position, cariogenic diet, prolonged breastfeeding, presence of dental plaque and hypoplasia. Logistic regression models were used to predict caries prevalence and negative binomial regression for caries experience. The prevalence of ECC was 29.1%, and the most frequent child temperament was ‘effortful control’. Regression model analyses, adjusting for covariates, showed no evidence of an association between any domain of children’s temperament (surgency, negative affect and effortful control) with the prevalence of caries or caries experience. This cross-sectional study found no association between childhood temperament and ECC in preschool children for this population. However, due to the specificity of this population, the association cannot be entirely ruled out. Further studies are needed to help understand the association between temperament and oral health, including the influences of family environment factors and culture.

## 1. Introduction

In many countries over the last 25 years, there has been a significant decline in the prevalence of Early Childhood Caries (ECC) [1]. Despite this, certain groups, such as children living in developing countries and unprotected subgroups in industrialized nations (for example, immigrants, ethnic minorities and rural communities), continue to have high prevalence of ECC [2,3,4,5]. ECC is defined as “the presence of one or more decayed (non-cavitated or cavitated lesions), missing (due to caries) or filled tooth surfaces in any primary tooth in a child under the age of six” [4]. ECC is a worldwide public health problem and a significant challenge for the dental profession. According to the 2017 Global Burden of Diseases study, over 500 million children suffered from caries in primary teeth [5]. This highly prevalent condition also carries high social, economic and medical costs [6].

The etiology of ECC is multifactorial, and factors such as low socioeconomic position, presence of visible plaque on teeth, presence of hypoplasia, frequent between-meal exposure to sugary snacks or drinks and prolonged breastfeeding (beyond 12 months) have been strongly associated with this condition [7,8].

Fisher-Owens’ conceptual model identified the determinants of ECC in three groups: the community, the family and the child [9]. Macrostructural determinants at the community level play an essential role in shaping family and child determinants. However, intermediate factors, such as the psychological characteristics of the family and children, may also influence the development of ECC. Parents may adopt certain practices according to the temperament of the child. For example, parents that regard their children as ‘difficult’ might try to calm them down by giving them sweet snacks or drinks [10]. In addition, children with more impulsive personalities could be more likely to consume more sugars [11]. However, most research on the risk factors of ECC have focused on the biological characteristics of the child and their eating behaviours, whereas children’s psychological characteristics are comparatively underexplored.

Child temperament refers to the initial state from which personality develops, linking behavioural differences to underlying neural networks [12]. It is relatively stable across time and modifiable by later environmental influences [13]. There is some evidence in the literature of an association between child temperament and dental caries. Although studies on this topic are few, it has been observed that some temperament traits such as emotionality, activity, sociability, shyness or impulsivity are associated with higher ECC risk [14,15,16,17,18,19]. These traits have also been associated with riskier behaviours such as no tooth-brushing before bedtime and poor feeding practices [20,21]. On the other hand, children with less challenging temperament traits tend to have parents who are less likely to give them sweetened drinks and are more likely to be caries-free [20].

Nevertheless, available evidence on the association between child temperament and ECC is conflicting. Not all temperament subgroups have been found to be associated with ECC. Furthermore, some studies have found no association between ECC and child temperament [14,22]. These differences in the association between ECC and temperament can be caused by wider macrostructural social determinants such as the influence of the cultural background on the social acceptability of child temperament and parental behaviours. Thus, studying different populations and social and cultural contexts is essential to better understanding the role of child temperament in the risk of ECC. So far, studies on ECC and temperament have been conducted in North America [14,15,16]; Thailand [17]; China [18]; India [19]; Iran [20,22]. This gap in current knowledge highlights the need to explore this relationship among other populations. Consequently, this study aimed to describe the dental caries experience of a sample of preschool children in Chile and to assess the association between early childhood caries and child temperament.

## 2. Materials and Methods

This was a cross-sectional study with participants recruited from preschool children who attended creches and kindergartens of the INTEGRA Foundation in the city of Temuco in southern Chile. INTEGRA is a public provider of early childhood education and care, and it is free-of-charge to families of low socioeconomic backgrounds. After approval from the ethics committee of the Universidad de La Frontera (Folio N° 020_17), all parents (*n* = 673) of children aged 3–5 years attending INTEGRA pre-schools received an open invitation to participate in this study. In addition, authorization by INTEGRA was given to access its creches and kindergartens. Informed consent was obtained from all subjects involved in the study prior to conducting a clinical examination and participating in the questionnaires. The children of parents who agreed to participate in the study by signing a consent form were included. Children with systemic diseases, those taking medication for chronic conditions or those with severe disabilities were excluded. Data were collected between October 2017 and November 2018. This manuscript was prepared following the STROBE recommendations [23].

The sample size was estimated according to the minimum requirements for multiple logistic regression analysis, as recommended by Peduzzi [24]. As the reported global prevalence of early childhood caries is 48% [25], 167 parent-preschool child pairs were required to perform a multiple logistic regression analysis yielding a power of 0.80, at a two-sided significance level of 0.05, using up to 8 independent variables. Sample size calculations were carried out using a sample size calculator for multiple logistic regression analysis [26].

Children underwent an oral examination. Clinical examination was carried out by four fully qualified dental researchers (i.e., dental surgeons) and previously calibrated in adapted rooms inside educational establishments. The calibration process consisted of a theoretical stage in which the examination teams (examiner and recorder) received theoretical training on the study protocol and how to complete the clinical record and perform the dental examination. Subsequently, a group of 15 children was examined to test the inter-examiner agreement. Two weeks later, they were examined again to determine the intra-examiner agreement. Inter-observer and intra-observers’ kappa statistics were higher than 0.87, which indicates an almost perfect level of agreement on the diagnosis of dental caries, according to Landis and Koch’s criteria [27]. At each site, examinations were conducted in a dedicated room adapted for this purpose using a portable LED light. Visual inspection of the mouth was done with a dental mirror and probe to remove detritus. Clinical data were recorded using a tooth level, following the World Health Organisation’s criteria and recommendations for oral health data collection; therefore, only cavitated carious lesions were recorded. The lesion was recorded as cavitated when the pit, fissure or smooth surface of the tooth had an unmistakable cavity, undercut enamel or soft tissue. [28]. Radiographic examinations were not performed, and teeth were not dried before scoring. Clinical data collection also included the assessment of hypoplasia and the presence of detritus. At the end of the dental examination, all children were given a toothbrush, and their parents received a report on the oral health status of their children with tailored recommendations for their child. As part of the preschool routine, children brush their teeth twice a day with 1000 ppm. fluoride toothpaste under the supervision of the kindergarten assistant.

Parents were asked to complete a questionnaire consisting of four sections: (i) sociodemographic information; (ii) oral health-related habits; (iii) the Very Short Form of the Children’s Behaviour Questionnaire (CBQ-VSF). The CBQ-VSF instrument consists of 36 questions to measure the three higher-order temperament domains of surgency, negative affection and effortful control. The instrument is considered a valid assessment of children’s temperament. CBQ-VSF has demonstrated good psychometric properties [29] and has been validated for use in early childhood [30]; (iv) Parents were also asked to complete a diet questionnaire by Lipari and Andrade [31]. This questionnaire evaluates types of food, their frequency, and their occasions of consumption. A cariogenic diet score was determined using the diet diary. The scale classifies the cariogenic diet as “Low”, “Intermediate” or “High” cariogenic risk.

The outcomes assessed were caries prevalence (children with at least 1 decayed tooth) and caries experience in the primary dentition (dmft scores). The clinical data used in this analysis included decayed teeth (dt), filled teeth (ft) and number of teeth present.

The exposure in this study included the three higher-order temperament domains as measured in the CBQ-VSF: surgency (tendency to act with impulsive and active behaviour); negative affect (predisposition to experience negative feelings and difficulty being soothed); and effortful control (voluntary regulation of attention and behaviour). Each domain has 12 items with scores ranging from 1 to 7 (Likert scale) with some items reverse-scored so that all items in a domain could be interpreted in the same direction. There is no total temperament score. Each domain score is calculated using the average of the answered items. Higher scores indicate that the child exhibited a temperament domain with more strength.

Covariates included in the analysis were selected because of their key role in ECC according to the literature [7,8]. These included:presence of dental plaque (Yes/No)presence of hypoplasia (Yes/No)cariogenic diet score classified as “Low”, “Intermediate” or “High” cariogenic risk [31]prolonged breastfeeding defined as breastfeeding for more than 12 months (Yes/No)socioeconomic position (SEP), which was determined by asking participants to self-classify within the four National Health Fund levels [32]. To these four groups, a “Private insurance” category was added. Health insurance depends directly on SEP. Public health insurance (FONASA) covers 70% of the population and focuses on lower-SEP individuals, whereas private insurance generally covers those in higher SEPs.

The statistical analysis provided basic descriptive information on sociodemographic, oral health status and children’s behaviour variables. For the statistical analysis, a bivariate analysis between each outcome and temperament domain was conducted, which was followed by a multivariate logistic regression model to predict caries prevalence and a negative binomial regression model to predict caries experience due to the data distribution of dmft showing overdispersion. The final full models included all three domains simultaneously plus covariates for each outcome. Data analyses were performed using STATA 16.0 (Stata Corp LP, USA).

## 3. Results

Of the 673 parents who were invited to participate in this study, 302 agreed to participate. Nonetheless, the final sample only includes 172 parent–child pairs with full records, achieving a completion rate of 57.0% and an overall response rate of 25.6%. The children’s mean age was 3.3 years (s.d. 0.6). Overall, there were more male (51.2%) than female participants. The majority of participants reported having public health insurance (91.8%), with the largest proportion (46.5%) in the low-income insurance group (Fonasa A).

Regarding the children’s socioeconomic position, the data on parental National Health Fund levels indicated that most children (67.4%) were in the “Very low” or “Low” income categories. Another 15.1% reported middle incomes. The remaining (17.1%) had higher levels of income. Table 1 shows demographic information and oral health-related habits.

Breastfeeding of longer than 12 months was reported by 14.0% of parents. By dietary risk, the largest proportion of the sample was in the intermediate-risk group (43.0%). Another 40.7% were in the high-risk group; the remainder were in the low-risk group (16.3%).

Regarding the children’s temperament, the mean score for the surgency domain was 4.5 (s.d. 0.9) and the negative affect had a mean score of 4.7 (s.d. 1.3), whereas effortful control had a slightly higher mean score of 5.6 (s.d. 0.9).

Dental caries experience ranged from 0 to 11 teeth, with 29.1% (n = 50) of children having at least one active dental caries (caries prevalence). The mean dmft score was 1.1 (s.d. 2.2) teeth. The majority of children (54.1%) presented visible dental plaque. Enamel hypoplasia was present in 16.3% of children (*n* = 28) (See Table 1). Except for the presence of visible plaque on the dmft score (*p* < 0.02), none of the other covariates had a statistically significant influence on either dmft score or dental caries prevalence.

Bivariate regression analyses did not show enough evidence of an association between any domain of children’s temperament (i.e., surgency, negative affect or effortful control) and the prevalence of caries or caries experience (Table 2).

This was also the case after running the models with all three temperament domains, adjusting for covariates (socioeconomic position, presence of hypoplasia and detritus and diet and breastfeeding) as shown in Table 2. Confidence intervals were wide and did not support the rejection of the null hypothesis of no association of caries prevalence and surgency (OR = 1.09, 95% CI = 0.64–1.88), negative affect (OR = 0.90, 95% CI = 0.64–1.27) or effortful control (OR = 0.83, 95% CI = 0.50–1.37), or of caries experience and surgency (RR = 0.93, 95% CI = 0.52–1.62), negative affect (RR = 0.91, 95% CI = 0.70–1.26) or effortful control (RR = 0.83, 95% CI = 0.47–1.47).

## 4. Discussion

The aim of this study was to assess whether child temperament was associated with dental caries among a population of preschool children. The present findings suggest that child temperament is not associated with either dental caries prevalence or caries experience, after accounting for sociodemographic and oral health factors. Similar results have been found when assessing children from equivalent ages but in different social and cultural contexts to those of the present study—for example, Abedizadeh and collaborators [22] in Iran or Quinonez and collaborators [14] in Canada—when assessing a small sample of children using the Emotionality Activity Sociability (EAS) Temperament Survey for Children.

On the other hand, Quinonez and collaborators [15] observed that negative child temperament was associated with dental caries among 4-year-olds in the USA using the ECBQ-VSF tool. In addition, Chankanka and collaborators [17] observed in Thailand that three out of nine temperament traits measured when children were 12 months-old were associated with dental caries when they were 18 months of age. In the same manner, a study in India found an association between ECC and preschool children’s temperament in 4 of the 5 dimensions measured [19].

One issue of particular relevance is the variation in how to assess temperament [16], as there are different ways to define and categorise child temperament. Some previous studies examining this association did not use standardised measures [16], making it difficult to compare their results, especially considering the difficulty in defining complex psychological concepts such as temperament. Among the strengths of this study is the use of the ECBQ-VSF scale, which is a validated tool to measure child temperament via parents’ perceptions [29,33]. This questionnaire has clear definitions of the main three domains of child temperament, allowing comparisons between studies.

Evidence shows that a child’s personality and temperament can contribute to the prediction of parenting behaviours [13]. Children with difficult temperaments can increase mothers’ psychological distress, resulting in ineffective parenting strategies [34]. This, associated with other familiar (e.g., parental stress, maternal health belief or mother’s low educational level) and community factors (e.g., low SES or access to dental care health), can result in harming the child’s health behaviours [35]. Since the association between temperament and dental caries is not direct, their relationship should be investigated in relation to other family and community factors.

Moreover, the lack of association observed in this study can be related to the cross-sectional data in the present analysis. Additionally, the contextual and cultural characteristics of this particular population as well as the multiple factors playing a role in ECC development may also have influenced the present results. Thus, factors not included in the present analysis may have a stronger role than child temperament among this specific population. In addition, it cannot be ruled out that the characteristics of the sample may have introduced sources of bias and uncertainty into the statistical analysis. It is also important to consider the relatively small sample size in the present study. Although it was large enough for the proposed analyses, the sample size did not have adequate power to model more complex mechanisms such as Fisher-Owens and collaborators’ framework [9]. Another possible limitation of this study is that the dental caries diagnosis was based on the WHO criteria without considering non-cavitated lesions, which can lead to underdiagnosis of the disease.

Nonetheless, the present study did adjust the regression models to include the most reported factors associated with EEC, which were identified in systematic reviews. In addition, the very specific characteristics of the population included in the study (i.e., children living in Temuco and going to INTEGRA preschools), may make the results less applicable to other populations. However, and as stated, local cultural and social characteristics are likely to play a role in the management of children’s temperament by parents, and thus, it is valuable to study different populations with different cultural and sociodemographic characteristics to assess how child temperament may impact the risk of ECC.

Preschool oral health is often overlooked in Chile. Additionally, there is no oral health program specifically designed to identify populations at risk of ECC. Information from this study will make an important contribution to the detection, prevention and management of early childhood caries. Furthermore, the present study highlights the need to consider broader “upstream”, macro-structural determinants of health that shape individual risk behaviours, as recommended by the wider research literature, and how these might shape ECC distributions. This is highly important due to the prevalence and serious health consequences of this condition in terms of pain, decreased quality-of-life, self-esteem, speech difficulties, living a life free of infection and the overall health of the child [36]. If child temperament is a risk factor for ECC, then education and other health professionals may play a role in identifying and referring preschool children to proper care services; thus, this highlights the importance of the involvement of paediatric primary carers with oral healthcare professionals in the prevention of ECC. When planning oral healthcare strategies for preschool children, it is recommended to consider psychological characteristics and the identification of personality traits which might serve as predictive behaviours during dental visits [37,38] and of later psychopathology [39].

## 5. Conclusions

The present study described the effect of temperament as a risk factor for ECC, providing preliminary insights into this specific risk factor for ECC within a Chilean population. The study found no association between child temperament and dental caries prevalence or caries experience in preschool children. Although this may be the case for this specific population, the association cannot be entirely ruled out. This further emphasizes the need to evaluate whether there is an association by using a more heterogeneous sample and with a design aimed at elucidating the understanding of temperament in dental healthcare settings, including the influences of community and of cultural environment factors.

## Figures and Tables

**Table 1 ijerph-20-03251-t001:** Demographics and oral health status and related behaviours.

Variable	Mean (SD)
dmft ^1^	1.09 (2.22)
**Variable**	***n* (%)**
Caries Prevalence	
Yes	50 (29.1)
No	122 (70.9)
Health Insurance ^2^	
FONASA A	80 (46.5)
FONASA B	36 (20.9)
FONASA C	36 (15.1)
FONASA D	16 (9.3)
Private Insurance	14 (8.2)
Presence of Detritus	
Yes	93 (54.1)
No	79 (45.9)
Hypoplasia	
Yes	28 (16.3)
No	144 (83.7)
Prolonged Breastfeeding	
Yes	24 (14.0)
No	148 (86.0)
Cariogenic Diet	
Low risk	28 (16.3)
Medium risk	74 (43.0)
High risk	70 (40.7)

^1^ dmft; decayed, missing, filled teeth in temporary dentition. ^2^ From lower (FONASA A) to higher income (Private Insurance).

**Table 2 ijerph-20-03251-t002:** Multiple regression models for temperament domains and presence of caries and dmft scores (*n* = 172).

	Caries Prevalence ^1^ OR (95% CI)	dmft ^2,3^ RR (95% CI)
Temperament Domain		
Surgency	1.09 (0.64–1.88)	0.93 (0.52–1.62)
Negative Affect	0.90 (0.64–1.27)	0.91 (0.70–1.26)
Effortful Control	0.83 (0.50–1.37)	0.83 (0.47–1.47)
Health insurance ^4^		
FONASA A	1	1
FONASA B	0.51 (0.19–1.34)	0.99 (0.32–3.00)
FONASA C	1.06 (0.40–2.82)	0.95 (0.30–2.96)
FONASA D	0.28 (0.05–1.39)	0.26 (0.05–1.33)
Private insurance	0.47 (0.11–1.96)	0.30 (0.06–1.43)
Hypoplasia	2.02 (0.81–4.99)	1.36 (0.47–3.99)
Presence of detritus	1.47 (0.73–2.98)	2.10 (0.91–4.88)
Prolonged breastfeeding	1.00 (0.37–2.70)	0.79 (0.25–2.49)
Cariogenic diet		
Low risk	1	1
Medium risk	0.42 (0.16–1.11)	0.68 (0.22–2.11)
High risk	0.54 (0.21–1.41)	0.99 (0.32–3.05)

^1^ Logistic regression (OR: odds ratio); ^2^ dmft; decayed, missing, filled teeth in temporary dentition; ^3^ Negative binomial regression (RR: rates ratio); ^4^ From lower (FONASA A) to higher income (Private Insurance).

## Data Availability

The datasets used and/or analysed during the current study are available from the corresponding author upon reasonable request.

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
