# Peer review of "Absence of Association between Child Temperament and Early Childhood Caries: A Cross-Sectional Study"

_ijerph, 2023, doi:10.3390/ijerph20043251_

Round 1

Reviewer 1 Report

Overall, I think this is an interesting manuscript, and that it is important to publish negative results. However, there is some room for improvement:

Abstract:

“it is suggested that psychological characteristics be considered when planning oral health prevention strategies.” – this conclusion cannot be derived from the studies results. Although it may be included in the discussion part of the manuscript, it should thus not appear as a conclusion

The ECC prevalence is stated as 70.9%, however, this does not correspond at all to the reported results in the results section???

Introduction:

l. 75: “Nonetheless” – I think this should be replaced by “ So far,” to support the argument

Methods:

Why were oral hygiene habits (e.g. frequency of tooth brushing) and the use of fluoridated toothpaste not included as covariates? Both are very important in the development of ECC. Or were they included in any of the questionnaires? If yes, please elaborate in the manuscript. If not, please add this as an important limitation.

Results:

I cannot follow the calculation of response rates. If 302 agreed to participate, and 172 were included, why were 130 persons excluded? 172/302 is not 44.9%, and the response rate should be calculated given the number of persons participating/ number of persons invited. Please revisit these numbers and add the necessary information.

Did any of the other covariates turn out to have a significant influence on dmft or caries prevalence in the bivariate or multivariate analysis?

Discussion:

As described in the introduction section, a difficult child temperament may lead to ECC via dietary changes (i.e. parents giving more sweets) or worse brushing habits etc. It is thus clear that the underlying mechanism can only be indirect, bad temper cannot directly cause caries. This is important to discuss, and may be one of the reasons why a potential association could not be detected.

Conclusion:

Please also see my comment regarding the conclusion in the abstract. Many of the arguments made in the conclusion are valid points for discussion, but should be moved to the discussion section. The conclusion focusses on the importance of temperament in ECC development although this is not at all supported by the study’s results. The conclusion should be limited to what can be derived from the results, with max. 1 or 2 sentences acknowledging that further research is still necessary/ it cannot be ruled out that temperament still plays a role.

Reviewer 2 Report

Title: ABSENCE OF ASSOCIATION BETWEEN CHILD TEMPERAMENT AND EARLY CHILDHOOD CARIES

MDPI_IJERPH_ 2170931- Reviewed 10th January 2023.

REVIEWER:

Comments and Suggestions for Authors. 

Authors present a cross-sectional study to assess the association between child´s temperament and ECC in 172 Chilean pre-schoolers (aged 3 to 5 years old) children. Child´s temperament was assessed on parent´s responses to EC Behaviour questionnaire; Outcome on dental caries prevalence and experience were done according to WHO criteria (dmft scores)

The aim is relevant for preventive policy needs in oral healthcare fields. The manuscript is some clear, relevant for the field. The cited references are mainly within the last 5 years.

Moderate editing of English language and style is required; 

Reference list required review and format to ACS style guide. Please see instructions for authors. Some pages of references also are in default. 

Title- Recommendation to add the type of study conducted in order to not be un affirmative conclusion. Please improve the title.

Abstract – Please improve the text. Recommendation to add the sentence “approval protocol by the ethics committee”, and the “informed consent obtained from parents for conducting clinical examination and participle in questionnaire.”

Recommendation to add the clinical criteria for dental caries diagnosis.

Other Permissions, such us for example, taken from the responsible of pre-schools should be refereed, supposed described in the approval protocol by ethics committee.

Main text, materials and methos, results , discussion and conclusions:

 Lines, 58 to 60,  and 64 a 67– Please we-write and correct; Add a new reference if possible – do not repeat the reference! Avoid to apply the same reference in  sequence. 

Lines 69 and 70- please re-write the sentence: “Some studies…” or add the missing references on “association between ECC and child temperament. 

Line 85 to line 88 – Please clarify how was done the screening ? All parents of children attending to Integra?

Line 88 – Please apply the number “(N=673)” near adequate sentence …supposed to be “parents of children”. Improve English writing.

Suggest to add a Flow Chart with screening, and how children were enrolled.

Line 101- Please describe the researchers profession, Medicines? Dentists? Other oral healthcare professionals? In short paragraph clarify and describe the how was done the “previously calibrated”. 

Suggest to indicate the calibration procedure of WHO Criteria and dfmt index/score. 

Line 119. – Clarify or add reference to “diet diary”- how was performed. Please see also line 137 to 139.

Line 199- Please substitute “sample” for “population. 

Line 204 and 206- Please remove “her”

Discussion chapter should be improved. Please compare and review the sentences on lines 199to 201, with the sentence described in line 253. 

Lines 261 to 266 – suggestion to move this to discussion chapter. 

Round 2

Reviewer 1 Report

I believe the manuscript is now suitable for publication.